rsos.royalsocietypublishing.org

computational mechanics/computer modelling and simulation

model validation, relative error, computational modelling, orthogonal decomposition

**Author for correspondence:**
Eann A. Patterson
e-mail: eann.patterson@liverpool.ac.uk

# A probabilistic metric for the validation of computational models

Ksenija Dvurecenska[1], Steve Graham[2], Edoardo Patelli[1] and Eann A. Patterson[1]

[1]School of Engineering, University of Liverpool, Liverpool L69 3GH, UK
[2]National Nuclear Laboratory, Chadwick House, Warrington WA3 6AE, UK

 EAP, 0000-0003-4397-2160

A new validation metric is proposed that combines the use of a threshold based on the uncertainty in the measurement data with a normalized relative error, and that is robust in the presence of large variations in the data. The outcome from the metric is the probability that a model's predictions are representative of the real world based on the specific conditions and confidence level pertaining to the experiment from which the measurements were acquired. Relative error metrics are traditionally designed for use with a series of data values, but orthogonal decomposition has been employed to reduce the dimensionality of data matrices to feature vectors so that the metric can be applied to fields of data. Three previously published case studies are employed to demonstrate the efficacy of this quantitative approach to the validation process in the discipline of structural analysis, for which historical data were available; however, the concept could be applied to a wide range of disciplines and sectors where modelling and simulation play a pivotal role.

## 1. Introduction

Computational models are widely used to evaluate and predict the future behaviour of engineering systems. Recent increases in computational capabilities have made it possible to simulate a large variety of processes. For instance, simulations are used to understand the mechanical behaviour of novel materials and to develop and optimize sustainable designs for engineering structures. The results from a simulation are nearly always used to inform decisions that are likely to have socio-economic and/or human consequences. In most cases, the modeller will not and, it has been argued philosophically [1], should not be the decision-maker which implies that the credibility of the results or predictions from the model becomes vital and can be enhanced through a verification and validation (V&V) process [2].

rsos.royalsocietypublishing.org    R. Soc. open sci. 5: 180687

Verification[1] can be summarized as ensuring that the mathematics of the model is being solved correctly, whereas validation[2] is establishing a level of confidence in a model as an accurate and reliable representation of the reality of interest.

From these definitions, it can be seen that verification of the model should precede validation and usually it is a process undertaken by the purveyors of commercial and academic software packages using verification benchmarks [4,5]. In this study, the focus has been on the validation process which is usually undertaken by a modeller who is using a verified software package. Initial discussions about computational model validation appeared in the literature during the second half of the twentieth century and coincided with the advent of simulation and modelling techniques that were enabled by the availability of computing power. Fishman & Kiviat [6] and Van Horn [7] were among the first to consider the idea of validation, and related questions, in the context of models in economics science, but their ideas are relevant to simulations in many areas of science and technology. They identified that a computational model is usually developed with particular objectives that reflect the intended use; and consequently, the simulation results have to be evaluated against these objectives. Sargent [8] added further specificity by including the term 'for the intended use' in the definition of model validation. The concept of model validation emerged during the 1980s [9–11] as being the comparison of model behaviour with the behaviour of a real system when both the simulation and observations are conducted under identical conditions; and it was consolidated into two guides for engineers, namely the AIAA guide for computational fluid dynamics simulations [12] and the ASME guide for computational solid mechanics models [3] in 1998 and 2006, respectively. These guides provide concise definitions and a generalized methodology for performing verification and validation, but neither include definitive step-by-step procedures.

A common approach to validation is to divide the available empirical dataset into a calibration or training subset and a validation subset, then to 'tune' the model parameters using the calibration subset only before testing the model predictions with the validation subset and then repeating the entire process with a different division of the dataset. There are some concerns about the dual use of data, or double-counting, involved in such an approach. However, Steele & Werndl [13] have argued this practice of double-counting is legitimate within a Bayesian framework, such as used recently for a linear regression model of the strength of composite laminates containing manufacturing defects [14]. In this example, the prediction uncertainty of the model was estimated using leave-one-out cross-validation [15]. When large datasets are unavailable for calibration and validation and/or the model has multiple input and/or output parameters, such as when modelling the spatial distribution of mechanical strain in an engineering structure over time, then a different approach is required. Recently, a CEN workshop agreement [16] has provided a detailed methodology for performing validations of computational solid mechanics models.

In solid mechanics, it has been common practice to validate numerical models using single data points, for example the maximum and minimum values of a response measured by a strain gauge. However, recent work has extended this approach to using fields of data acquired from optical measurement techniques [17], e.g. stereoscopic digital image correlation. In these circumstances, the measured and predicted data fields are rarely in the same coordinate system or have the same data pitch, orientation or perspective and this renders direct comparisons problematic. Patterson and his co-workers have represented both measured and predicted data fields as images in order to apply orthogonal decomposition techniques [18] and enable straightforward comparison for the purpose of validation [19] as well as model updating [20]. In image decomposition, a set of polynomials are used to represent the image such that only the moments or coefficients of the polynomials are required to describe the image [21]. Image decomposition using orthogonal moments not only reduces the dimensionality of the data from an image matrix to a feature vector but is also invariant to rotation, scale and translation of the images [19]. Clearly, it is important to ensure that a feature vector is a good representation of the original data before using it in a validation procedure, and the CEN guide [16] recommends that the reconstructed image must satisfy the criteria that the uncertainty introduced by the decomposition process, $u_{\text{deco}}$ must be less than the minimum measurement uncertainty in the measured dataset, $u_{\text{cal}}$ and that there should be no cluster of points in the image where the residual is greater than three times the decomposition uncertainty, $u_{\text{deco}}$. A cluster was defined as a region of adjacent pixels representing more than 0.3% of the data, and the minimum measurement uncertainty

---

[1]The ASME guide [3] defines verification formally as 'the process of determining that a computational model accurately represents the underlying mathematical model and solution'.

[2]The ASME guide [3] defines validation formally as 'the process of determining the degree to which a model is an accurate representation of the real world from the perspective of the intended uses of the model'.

rsos.royalsocietypublishing.org    R. Soc. open sci. **5**: 180687

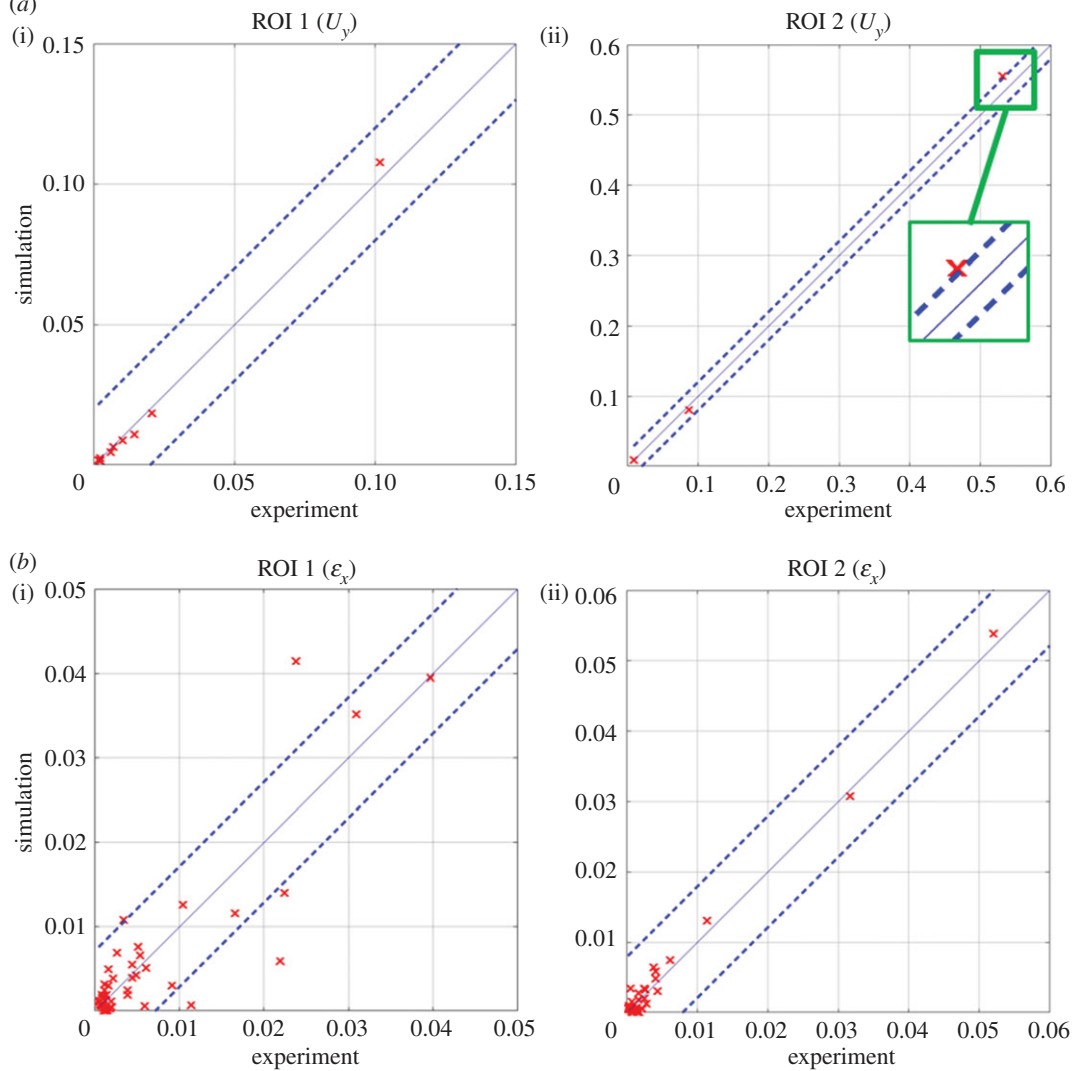

**Figure 1.** Graphical comparisons, using the approach recommended by the CEN guideline [16] for evaluating the acceptability of model predictions, of the Zernike moments representing the predicted (*y*-axis) and measured (*x*-axis) transverse displacement (*a*(i,ii)) and longitudinal strain (*b*(i,ii)) in regions 1 (*a*(i),*b*(i)) and 2 (*a*(ii),*b*(ii)) of the I-beam subject to three-point bending shown in figure 3 (based on Lampeas *et al.* [23]). The predictions can be considered acceptable when all of the data fall within the zone bounded by the broken lines that are defined by equation (1.2) based on the measurement uncertainty.

was obtained using a calibration procedure for the measurement apparatus [22], while the decomposition uncertainty was defined as follows:

$$u_{\text{deco}}^2 = \frac{1}{N} \sum_{i,j=1}^{N} (\hat{I}(i,j) - I(i,j))^2, \tag{1.1}$$

where $I(i,j)$ and $\hat{I}(i,j)$ are the original and reconstructed images, respectively, and $N$ corresponds to the number of data points in the images. When the images of the measured and predicted data fields have been decomposed using the same process, the resultant moments can be plotted against one another to provide a simple comparison, as illustrated in figure 1. The CEN guide recommends that the model can be considered valid if all of the moment pairs fall within the zone described by

$$S_P = S_M \pm 2u_{\text{exp}}, \tag{1.2}$$

where $S_P$ and $S_M$ are the moment values representing the predicted and measured data fields, respectively, and $u_{\text{exp}}$ is the total uncertainty in the measured data, which is given by

$$u_{\text{exp}} = \sqrt{u_{\text{cal}}^2 + u_{\text{deco}}^2}. \tag{1.3}$$

rsos.royalsocietypublishing.org R. Soc. open sci. 5: 180687

Although the CEN guide [16] was prepared from the perspective of solid mechanics and hence the predicted and measured data are in the form of displacement and strain fields, because the decomposition process is applied to images of the data fields, it could be used for any application in which predicted and measured data fields can be treated as images. The approach results in a statement about the adequacy of the representation of reality by the model but does not provide information about the degree to which the predictions represent the measurements.

## 2. Validation metrics

Although the validation of simulation results is often referred to as a single process, at a more detailed scale it can be divided into two activities [24]: first, the difference between the predicted and measured results is computed with the aid of a statistical comparison; and second, the outcome is evaluated in the context of the adequacy requirements. The statistical comparison is usually expressed in the form of a validation metric, i.e. a function representing the distance between the two results in the appropriate domain [25]. An ideal validation metric should be quantitative, objective and include a measure of the uncertainty in the measured and predicted results [3,24,26,27]. Berger and Bayarri [28] have suggested that validation methodologies can be classified as either frequentist or Bayesian; however, the approach recommended in the CEN guide is a form of hypothesis testing that provides a Boolean result, i.e. the model is either acceptable or unacceptable, without any indication of the quality of the results from the model. In some instances, particularly when the model has been found to be unacceptable, without information about the quality of the predictions, decision-makers will be unable to identify an efficient trade-off for the next set of actions, apart from a general decision to refine the model [3]. This information gap has been closed in this study by integrating a Boolean decision, based on the CEN approach, with a quantitative validation metric that is frequentist in nature.

In the frequentist approach, the measured data are assumed to be true and used to compute a relative error in the predicted data, i.e. the difference between the response of the model and the experiment. In reality, the measured data cannot be considered as absolutely true because there will be uncertainties and errors associated with the measurements and these should be accounted for when evaluating the discrepancy between the measured and predicted datasets [26]. Oberkampf and Barone [24] calculated both an average and a maximum relative error and then estimated confidence intervals for both relative errors, which allowed the degree of validity to be expressed. Their work is often cited, e.g. [29–31], both for its summary of the validation procedure and its definition of a validation metric; however, the metric is often simplified [32,33] because it is not robust when a system response cannot be time-averaged or is close to zero-valued. Kat & Els [34] avoided these issues by evaluating the absolute percentage relative error of each pair of data values and comparing it to a specified threshold set by the accuracy requirements, which allowed them to provide a probability of the predictions from the model being within the specified threshold. They assumed that the data were deterministic quantities and did not include an uncertainty analysis. Bayesian analysis permits uncertainty to be considered but does not appear to have been used to produce a statement about the validity of a model, i.e. to quantify the degree to which predictions are a reliable representation of reality. Instead, most reports in the literature on this topic are associated with model calibration or updating [30,31,35], which is the process of adjusting model parameters to reduce the discrepancy between predictions and a specified benchmark. In Bayesian analysis, initial information about the quantity of interest is described by a probability distribution, known as a prior distribution, and is updated using additional information described in a probability distribution, known as a likelihood, to produce a new or updated probability distribution describing the quantity of interest, known as the posterior distribution.

The ratio of the prior and posterior distributions is known as the Bayes factor, which both Rebba & Mahadevan [36] and Liu et al. [27] have identified as a possible validation metric together with an associated confidence index. While it might be possible to use an uninformed, naive prior derived from theory, in general, the choice of the prior distribution and the data to be included in the likelihood is subjective [37] which is inappropriate for an objective validation metric.

## 3. Developing a new validation metric

The motivation for developing a new validation metric was to advance the approach recommended in the CEN guide [16]. A new probabilistic metric, which is applicable to data fields, is proposed to include a measure of the extent to which predicted data are representative of reality as described by

rsos.royalsocietypublishing.org    R. Soc. open sci. **5**: 180687

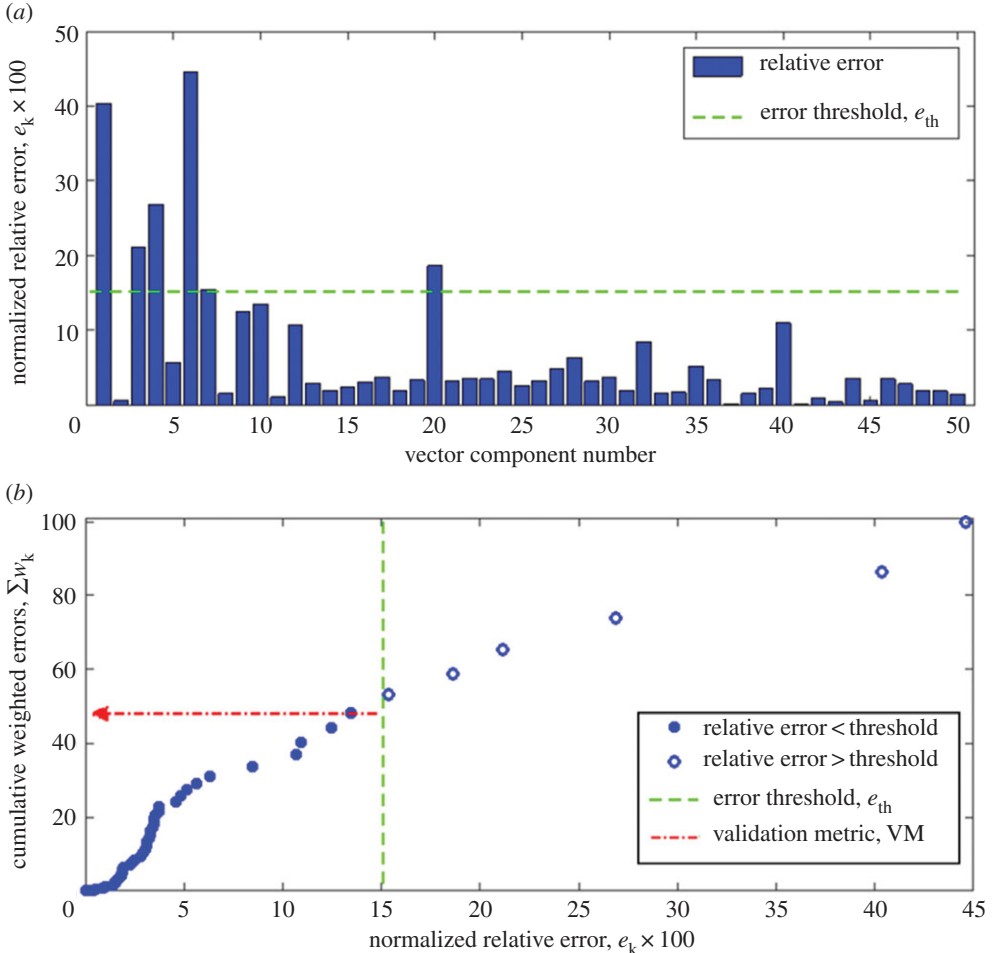

**Figure 2.** A bar chart of normalized relative errors (*a*) based on equation (3.1) and multiplied by 100 to allow the error threshold from equation (3.3) to be shown; and the cumulative distribution (*b*) of ranked weighted errors computed using equation (3.2) for the predicted longitudinal strain field in region 1 of the of the I-beam subject to three-point bending shown in figure 3; based on equation (3.4) the validation metric is the sum of those errors below the threshold, i.e. the filled symbols in (*b*).

measured data. The predicted and measured data are represented by a pair of feature vectors, $S_P$ and $S_M$, respectively, obtained by orthogonal image decomposition following the process described by CEN [16]. The proposed validation metric is evaluated in four steps: (i) compute a normalized relative error, $e_k$ for each pair of vector components; (ii) compute a weight for each error, $w_k$; (iii) define an error threshold, $e_{th}$; and (iv) calculate the validation metric, VM as the sum of those weighted errors, $w_i$ less than the error threshold, $e_{th}$.

The normalized relative error is defined as

$$e_k = \left| \frac{S_{P_k} - S_{M_k}}{\max_{m \in S_M} |S_{M_m}|} \right|, \tag{3.1}$$

where $S_{P_k}$ and $S_{M_k}$ are the $k$th vector components representing the predicted and measured results, respectively, and $\max_{m \in S_M} |S_{M_m}|$ is the magnitude of the measurement vector component with the largest absolute value. A bar chart of a typical set of normalized relative errors, $e_k$ is shown in figure 2 for the longitudinal strain field in an I-beam subject to three-point bending. The weight, $w_k$ of each error is defined as its percentage of the sum of the errors, i.e.

$$w_k = \frac{e_k}{\sum_{k=1}^{n} e_k} \times 100, \tag{3.2}$$

where $n$ is the number of components in each vector. The error threshold, $e_{th}$ is calculated by combining the approaches employed by Kat & Els [34] and Sebastian *et al.* [19] and normalizing the expanded

uncertainty in the measurement data, i.e.

$$e_{th} = \frac{2u_{\exp}}{\max\limits_{m \in S_M} |S_{M_m}|} \times 100. \tag{3.3}$$

This error threshold has been evaluated for the data in figure 2 and shown as a dashed line. Once these three steps were completed, the weighted errors, $w_k$, were compared to the error threshold, $e_{th}$, and the sum of those errors less than the threshold computed to yield the validation metric, VM, i.e.

$$\text{VM} = \sum_i w_i ||_{w_k < e_{th}} \tag{3.4}$$

where $||$ is an indicator function which takes the value 1 when $w_k < e_{th}$ and otherwise has a value zero. This process is represented graphically in figure 2b by ranking the values in figure 2a and then calculating their cumulative weighted value. Following the interpretation of Kat & Els [34], this sum corresponds to the probability of the normalized errors being equal to or less than the experimental uncertainty. From the validation perspective, VM represents the probability that model is representative of reality for a specified intended use.

A minimum number of points are required to define the cumulative distribution, shown in figure 2b, in order for the validation metric to yield reliable results. It is impossible to define this minimum number of points for an unknown distribution; however, for the simplest nonlinear curve, i.e. a conic, at least five points are required assuming there is no uncertainty associated with the location of the points, according to Pascal's theorem [38]. Hence, it is reasonable to assume that $n_{\min} \geq 5$. In addition, the number of points in the cumulative distribution corresponds to the number of components in each of the feature vectors, $S_P$ and $S_M$, or the number of moments used in the orthogonal decomposition of the images of the predicted and measured data fields. For data fields in which the variable is a nonlinear function of both spatial coordinates, the orthogonal polynomials recommended in the CEN guide [16], i.e. Chebyshev and Zernike, will require at least six moments or shape descriptors to describe the data field. Data fields that are linear functions of the spatial coordinates can be compared using simpler approaches than proposed here, so that practically, $n_{\min} = 6$.

# 4. Case studies

The application of this new validation metric has been demonstrated for three case studies, which are described in this section, using previously published predictions and measurements, including two from a recent inter-laboratory study (or round-robin exercise) on validation [39]. In part, these case studies were chosen because the data were available and the minimum measurement uncertainties had been established following methodologies similar to that recommended by the CEN guide [16] and were relatively small. In each case, data fields from computational models and physical experiments were treated as images and post-processed using an identical orthogonal decomposition methodology, following Sebastian et al. [19], to produce feature vectors, $S_p$ and $S_M$.

## 4.1. I-beam subject to three-point bending

The data for this case study were taken from an earlier study [23] of the efficacy of the validation methodology described in the CEN guide [16]; hence, only brief details of the model and experiments are included here. A half-metre length of aluminium I-section with overall cross-section dimensions $42 \times 65$ mm was subject to static bending by a central load while supported symmetrically by two 50 mm diameter solid rods of circular cross-section that were 450 mm apart. The thickness of the web and flange was 2.5 mm and a series of four 35 mm diameter circular holes penetrated the web at 100 mm intervals along its length, as shown in figure 3a. In the experiment, a stereoscopic digital image correlation system was used to acquire displacement data and the minimum measurement uncertainty was established as 10 μm for displacement and 30 με for strain measurements using the calibration procedure described in [40]. A finite-element model was created using 23 135 shell elements with the Ansys software package and using an elasto-plastic material model with kinematic hardening. The predicted and measured data fields were decomposed using 400 Zernike moments, but only significant coefficients, i.e. lower terms of the polynomial that represent main features of the data field within the specified threshold [23], were included in the validation.

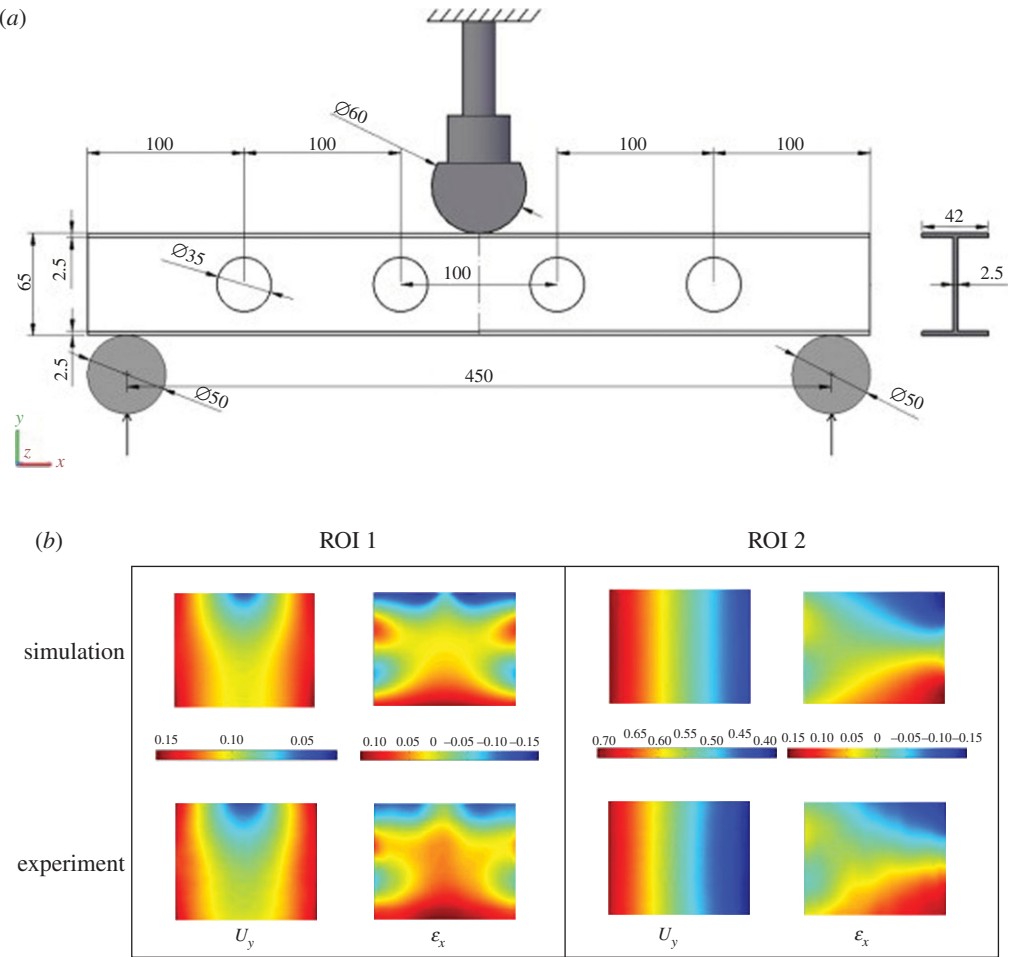

**Figure 3.** A diagram (*a*) of the I-beam subject to three-point bending showing the regions of data used in case study 1 (all dimensions in mm) together with the predicted and measured fields of transverse displacement (mm) and longitudinal strain (%) (*b*) (adapted from Lampeas *et al.* [23]).

**Table 1.** Case study 1: I-beam subject to three-point bending.

|  | $u_{exp}$ (%) | $e_{th}$ (%) | validation metric, VM (%) |
|---|---|---|---|
| region 1 |  |  |  |
| $u_y$ | 2.69 | 24.15 | 100 |
| $\varepsilon_x$ | 3.57 | 15.11 | 48 |
| region 2 |  |  |  |
| $\varepsilon_x$ | 3.97 | 11.53 | 100 |

In this case study, the extent to which the predictions represent the measurements of the transverse displacement of the web and the longitudinal strain in regions 1 and 2 in figure 3*a* were evaluated. The probability that the predictions are acceptable was found to be 100% and 48% for the transverse displacement and longitudinal strain, respectively, in region 1 while the corresponding probability in region 2 for the strain was 100%. These results are summarized in table 1 together with corresponding values of measurement uncertainty, $u_{exp}$ (from [23]) and error threshold, $e_{th}$ computed using expression (3.3). No value of the validation metric was calculated for the displacement in region 2 because less than six shape descriptors were required to represent the displacement field due to its simple shape, as shown in figure 3*b*.

These outcomes correlate well with those in figure 1 obtained by following the CEN guidelines. For example, a relatively low probability was found for the longitudinal strain, $\varepsilon_x$ in region 1, which corresponds to the widely scattered data points in figure 1*b*(i). Hence, it can be concluded that

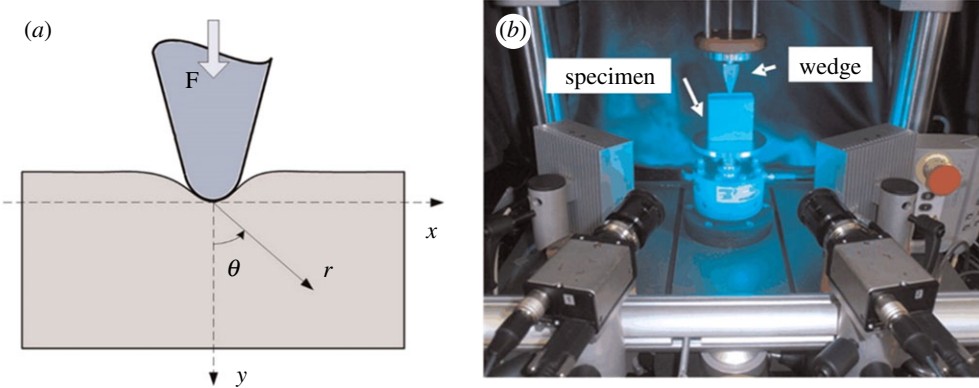

**Figure 4.** Schematic diagram (*a*) and photograph (*b*) of the indentation of a rubber block (60 × 60 × 30 mm) by a rigid indenter (adapted from Tan *et al.* [41]).

implementation of the relative error metric improves upon the binary outcome of the CEN methodology by quantifying the quality of the predictions.

## 4.2. Rubber block subject to indentation

The indentation of a 60 × 60 × 25 mm rubber block by a rigid wedge has been investigated previously by experiment and modelled analytically [41] and computationally [39]. Consequently, only a brief outline is provided here. Deformation data for the rubber block were acquired using a stereoscopic digital image correlation system when a compressive displacement load of 2 mm was applied across the entire 25 mm thickness of the block by an aluminium alloy wedge of external angle 73.45° and tip radius 1.68 mm (figure 4). The stereoscopic digital image correlation system was calibrated and found to have minimum measurement uncertainties of 3.2 and 23.8 μm for the in-plane [22] and out-of-plane [42] displacements, respectively. Predictions of the *x*-, *y*- and *z*-direction displacements were obtained from a finite-element model simulated in the Abaqus 6.11 software package using 49 920 three-dimensional eight-noded linear elements for the block and 2870 three-dimensional four-noded bilinear quadrilateral elements for the wedge. The material of the wedge was assumed to be rigid while the rubber was modelled as a hyperelastic material defined by the Mooney–Rivlin relationship with the constants taking the following values: $C_{10} = 0.9$ and $C_{01} = 0.3$ with a bulk modulus, $J = 20$.

The measured and predicted displacement fields are shown in figure 5 and were decomposed using Chebyshev polynomials. In order to achieve average reconstruction residuals that were just below the appropriate minimum measurement uncertainties, as recommended by the CEN guide [16], 170, 210 and 15 moments were employed to describe the surface displacement in *x*-, *y*- and *z*-directions, respectively. The values for validation metric, VM, for the *x*-, *y*- and *z*-direction displacements were 82.48%, 62.42% and 34.3%, respectively, based on error thresholds of 9.95%, 1.19% and 24.63% for the *x*-, *y*- and *z*-direction displacements.

These results correlate well with outcomes observed in figure 5, which were obtained by following the CEN methodology. As was expected from the visual comparison of the fields in figure 5, the model is quite poor at predicting displacement in the *z*-direction and, even given the high uncertainty, the value of VM is very low. At the same time, the probabilities for the predictions of displacements in *x*- and *y*-directions have been reflected successfully and the validation metric quantified the differences.

## 4.3. Bonnet liner impact

Burguete *et al.* [43] have described the analysis of the displacement fields for an automotive composite liner for a bonnet or hood subject to an impact; and so only an outline of the data acquisition and processing is given here. The composite liner, which had overall dimensions of approximately 1.5 × 0.65 × 0.03 m, was subject to a high velocity (70 m s$^{-1}$), low-energy (less than 300 J) impact by a 50 mm diameter projectile with a hemispherical head. A high-speed stereoscopic digital image correlation system was used to obtain maps of out-of-plane displacements at 0.2 ms increments for 100 ms. The minimum measurement uncertainty was established to be 14 μ$\varepsilon$ at 290 μ$\varepsilon$ rising to 29 μ$\varepsilon$

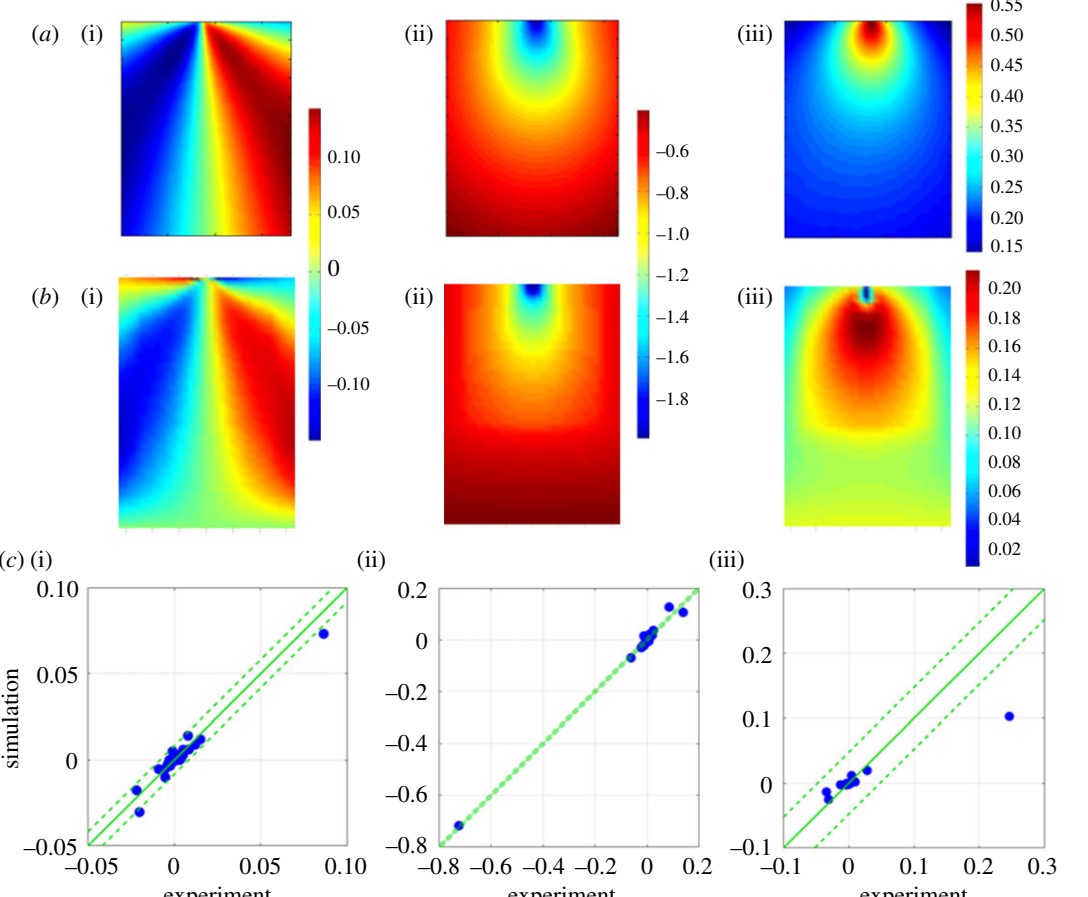

**Figure 5.** Measured (*a*) and predicted (*b*) *x*-direction (i), *y*-direction (ii) and *z*-direction (iii) displacement fields for a 28.5 × 23 mm area of the rubber block shown in figure 4 when it was subject to 2 mm displacement load by the wedge in the *y*-direction; and plots obtained using the CEN methodology [16] (*c*). The centre of the top edge of each data area corresponds to the location of contact by the wedge and the units are millimetres.

at 2110 µε [43]. The finite-element code Ansys-LS-Dyna was employed to model the bonnet liner following impact using an elastic-plastic material model with isotropic damage and four-noded elements based on a Belytschko–Tsay formulation. Typical fields of predicted and measured fields of out-of-plane displacements are shown in figure 6 and were decomposed using adaptive geometric moment descriptors (AGMD) specifically tailored for the complex geometry of the liner. Burguete *et al.* [43] compared the data fields from the model and experiment for 100 ms following impact by plotting the absolute difference between pairs of corresponding AGMDs as shown in figure 7*a*. They concluded that when any of the absolute differences were greater than the uncertainty in the experiments, indicated by the broken lines in figure 7*a*, then the model was not valid. In this study, the probability of the model being acceptable was assessed using the validation metric in equation (3.4) for each increment of time for which a displacement field was measured. The result is shown in figure 7*b* together with the result obtained by Burguete *et al.* [43]. The trends in acceptability implied by both plots in figure 7 are similar, with the predictions being a reasonable representation of the experiment for about 0.035 s after impact. Burguete *et al.* observed that, after this time instance, a crack developed in the test specimen unexpectedly which was not permitted to develop in the model and this accounts for the poor performance of the predictions.

## 5. Discussion

The proposed validation metric is based on a relative error metric, but through the application of appropriate normalizing of the relative error and the error threshold, the drawbacks of the previous frequentist approaches are avoided. This means that, unlike previous metrics, the proposed metric is

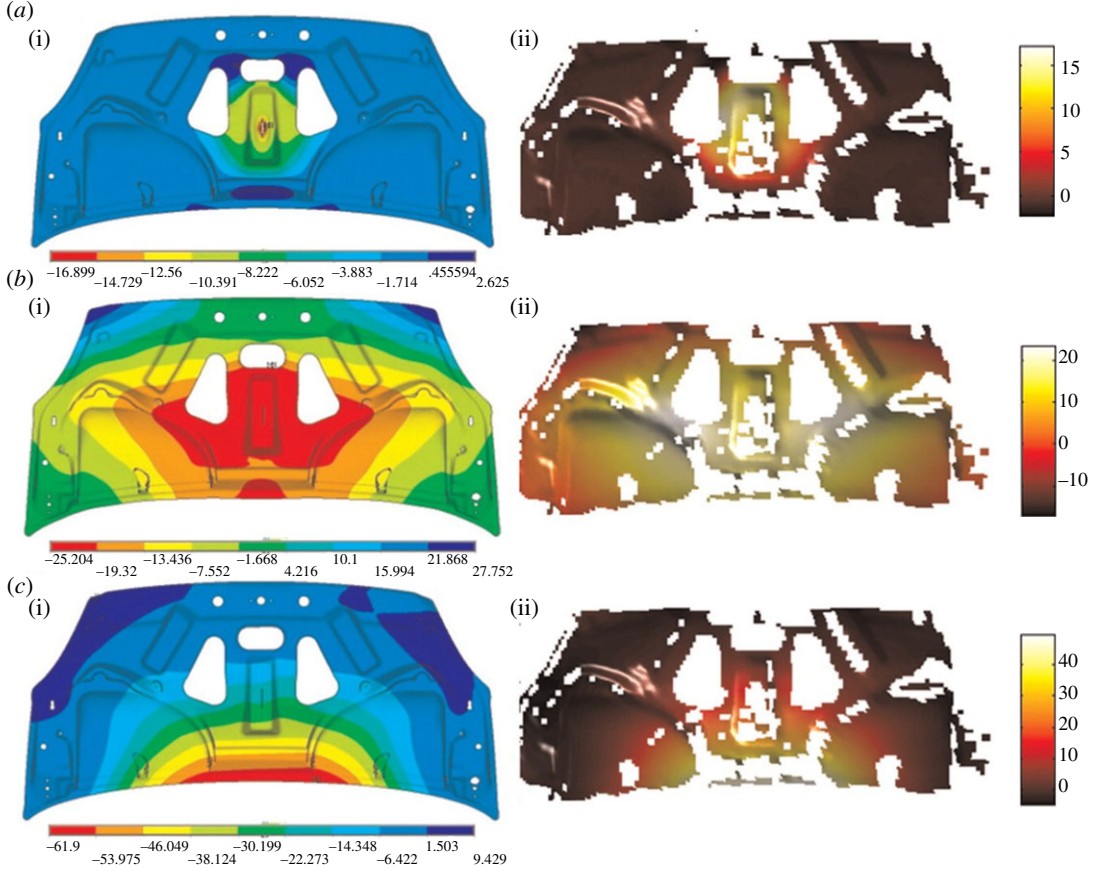

**Figure 6.** Predicted (*a*(i), *b*(i), *c*(i)) and measured (*a*(ii), *b*(ii), *c*(ii)) out-of-plane displacement fields for the automotive bonnet (hood) liner (approx. 1.5 × 0.65 × 0.03 m) at 40, 50 and 60 ms (from *a*(i,ii) to *c*(i,ii)) after a high-speed, low-energy impact by a projectile in the centre of the liner (adapted from Burguete *et al.* [43]).

capable of evaluating data with a naturally high variance between the individual values in the dataset, including very small values close to zero. It also takes into account uncertainties in the measurement data. In part, these advantages are a result of the choice of mean absolute percentage error as a basis for calculating the validation metric following the work of Kat & Els [34] and which allows the measurement uncertainty to be directly employed as an error threshold. This ease of interpretation and the direct proportionality of the influence of each contribution to the absolute value of error were additional reasons for the choice of mean absolute percentage error instead of a root mean square error. The result is a value for the probability that predictions from a model are a reliable representation of the measurements based on the uncertainty in the measurements used in the comparison. This allows the outcome of the validation process to be expressed in a clear quantitative statement that reflects the complete definition of the validation process. Such a statement should include the following three components:

— the probability of the model's predictions being representative of reality,
— for the stated intended use and conditions considered, and
— based on the quality of the measured data defined by its relative uncertainty.

For example, one of the validation outcomes for the rubber block case study can be expressed as follows: '*there is an 83% probability that the model is representative of reality, when simulating x-direction displacements induced by a 2 mm indentation, based on experimental data with 10% relative uncertainty*'. The implementation of this type of statement would represent a significant advance on current practice and could be interpreted relatively straightforwardly by decision-makers. The outcome of this type of modified validation process allows the decision-maker, e.g. customer or stakeholder, to make the final judgement based on the evidence from the validation and their required or desired level of quality. When the level of agreement between predicted and measured data is inadequate for the intended

rsos.royalsocietypublishing.org    R. Soc. open sci. **5**: 180687

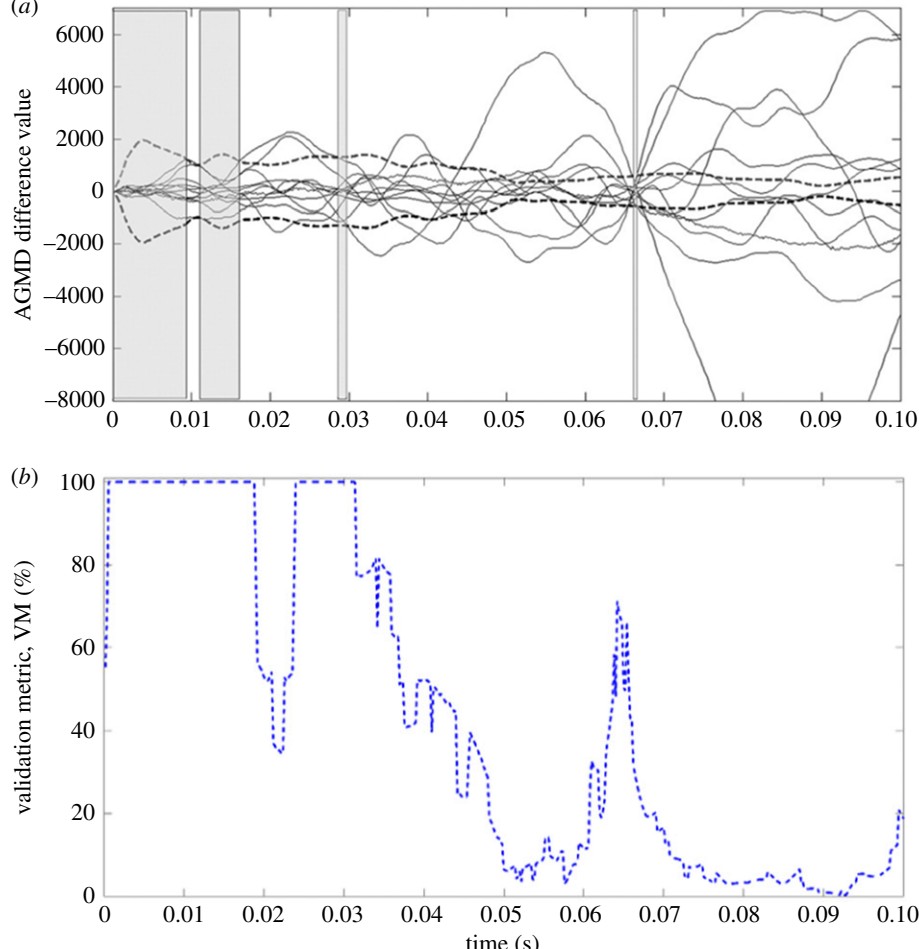

**Figure 7.** (a) Absolute difference between corresponding adaptive geometric moment descriptors describing the predicted and measured out-of-plane displacement field of the automotive bonnet liner during the 0.1 s following impact; and (b) the corresponding probability of the predictions being a reliable representation of the measurements based on incorporating the weighted relative error and error threshold into the validation metric, VM, using equation (3.4) ((a) is adapted from Burguete et al. [43]).

purposes of the model, then both ASME [3] and CEN [16] guides recommend that both the model and the experiment should be reviewed before repeating the validation process. The use of model updating techniques [20] might be appropriate at this stage.

Brynjarsdóttir & O'Hagan [44] have discussed the issue that experiments and simulations both mimic reality so that both have a certain level of approximation which has to be accounted for during a validation process. In particular, analysis that does not account for the discrepancies arising from these approximations may lead to biased and over-confident predictions. Hence, it is not enough to compare a simulation with an experiment, but also it is necessary to consider the relationship of the experiment to reality [45]. In other words, to recognize that the process of experiment design results in a representation of the real-life situation based on our current understanding and that the resultant measurements should not be regarded as the absolute truth. Of course, measurements made directly in the real-life situation are likely to be closer to the truth than those made using physical models, but the measurement process will always influence the measurement data leading to uncertainty about the truth. Hence, the last component of the statement above would ideally include information about the discrepancy between the truth and the measurements used in the validation process. However, this information is usually unavailable and, as a consequence, some caution, and awareness of context, needs to be exercised in employing the type of statement expressed above in italics; nevertheless, it represents a potential improvement on current practice in terms of its specificity.

The new validation metric, VM in equation (3.4), has been described in generic terms and the case studies illustrate its application to information-rich spatial data fields using feature vectors; however,

the vectors, $S_P$ and $S_M$ describing the predicted and measured data could be constructed from many types of data, including time-series data. There is an implicit assumption in the use of the orthogonal image decomposition process to compare data fields, which is of one-to-one correspondence between the components of the feature vectors representing the data fields. This could be viewed as a potential limitation of the approach because this correspondence might not be present when some decomposition processes are used; however, the decomposition process employed here and recommended in the CEN guide [16] was designed to deliver this correspondence. The measurement data in each of the case studies were displacement fields obtained using digital image correlation and were chosen based on the availability of both predicted and measured data fields and of measurement uncertainties. Digital image correlation has become almost ubiquitous in experimental mechanics and hence its use here; nevertheless, the decomposition technique is widely applicable and has been used for data fields from thermoelastic stress analysis and projection moiré [46]. The generic nature of the approach should allow its application in a wide variety of disciplines, for instance computational fluid dynamics, computational electromagnetics or landscape topography evolution modelling, and sectors, including civil, electrical and mechanical engineering, whenever the predicted and measured data are available as maps that can be treated as images. In some applications, it is not possible to generate measurement data at all points in the region of interest, such as when optical access is obstructed or only a small number of point sensors can be employed or when the system is inaccessible, for example in a nuclear power plant. In these circumstances, when there is a sparsity of data, the relative error cannot be calculated for all of the predictions and this shortfall should be reflected in the statement about the outcome of the validation process, i.e. it would be appropriate to state what percentage of the predictions were used in constructing the validation metric and how well the position of these data values covered the region of interest. The interpretation of this additional information will be specific to the intended use of the model and hence no prescription is provided here.

The three case studies are a progression from a linear elastic planar static analysis, through a large deformation elastic static analysis to a nonlinear elasto-plastic time-varying analysis. Although this progression provides increasing challenges to both modellers and experimentalists, all of these cases are mechanical systems that can be represented by deterministic models and for which it is possible to design and conduct repeatable experiments with relatively low levels of measurement uncertainty. Many analyses in engineering will fall within the same classification; however, in its current form, the validation metric cannot be applied to probabilistic models or to nonlinear dynamic models with solutions in state space that lie outside this classification.

The approach to the validation process described in the ASME V&V guide [3] implies that it should be an interactive effort between those responsible for the model and those developing and conducting the experiments required to generate measurement data. However, it is unlikely that either group will be responsible for making decisions based on the predictions from the model and hence the credibility of the model becomes a critical factor. Model credibility is the willingness of others to make decisions supported by the predictions from the model [47]. Thus, it is important to present the outcomes from the validation process in a manner that can be readily appreciated by decision-makers who may not be familiar with principles embedded in the model or the approach taken to validation, including the techniques used to acquire the measurement data used in the validation process. Patterson & Whelan [48], in the context of computational biology, have discussed strategies for establishing model credibility, including incorporating a high degree of transparency and traceability in the validation process, recognizing the inadequacy of experiments as representations of the real world, stating the uncertainties associated with the data in the outputs from the validation process, and expressing the accuracy of the representation of the real world in terms of probabilities. The new validation metric combined with the proposed statement about the outcome of the validation process addresses these last two issues.

# 6. Conclusion

A new validation metric based on a frequentist approach has been proposed. The advantages of the metric are that it can handle datasets with large amplitude variations in data values as well as close-to-zero values and that the uncertainty in the measured data is also included in the metric. When it is combined with an appropriate orthogonal decomposition technique, then the dimensionality of large matrices of data can be reduced to feature vectors that enable data-rich maps of measurements to be used in the validation of corresponding predictions. The new validation metric allows a statement to

be constructed about the probability that the predictions from a model represent reality based on experimental data with a given relative uncertainty for a specified intended purpose.

Three case studies have demonstrated the use of the new metric in computational mechanics for a linear elastic planar static analysis, for a large deformation elastic static analysis, and for a nonlinear elasto-plastic time-varying analysis. The outcomes obtained with the new validation metric were more quantitative and informative than the previous validation procedures but qualitatively equivalent. Although these case studies relate to structural analysis, the principles illustrated are applicable to analysis in a wide range of fields including bioengineering, Earth sciences and nuclear engineering.

Finally, it is proposed that the new metric can be used to construct a clear quantitative validation statement about a model that contains three core components: (i) the probability that model's predictions are representative of reality; (ii) for the intended use and conditions for which the comparison with measurements was performed and (iii) the uncertainty in the measurement data.

Data accessibility. Our data are deposited at the Dryad Digital Repository: http://dx.doi.org/10.5061/dryad.2qp305p [49].
Authors' contributions. K.D. performed the research and prepared the first draft of the paper. E.A.P. conceived and supervised the study and prepared the final draft of the paper. S.G. and E.P. supervised the study and contributed to the interpretation of the results and refinement of the methodology, including the interpretation of the ranked weighted errors as a cumulative distribution function (E.P.). All authors gave final approval for publication.
Competing interests. We declare we have no competing interests.
Funding. K.D. was supported by a studentship funded by the UK Engineering and Physical Sciences Research Council and by the National Nuclear Laboratory. E.A.P. was in receipt of a Royal Society Wolfson Research Merit Award.
Acknowledgements. The authors are grateful to George Lampeas, Vasilis Pasialis, Xiaoshan Lin, Luis Felipe-Sese, Xiaohua Tan and Weizhong Wang for access to their data for the case studies.

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
