## [Reviewer comments · Royal Society Open Science]

Review History

RSOS-180687.R0 (Original submission)

Review form: Reviewer 1 (Martin Treiber)

Is the manuscript scientifically sound in its present form?

No

Are the interpretations and conclusions justified by the results?

Yes

Is the language acceptable?

Yes

Is it clear how to access all supporting data?

Yes

Do you have any ethical concerns with this paper?

No

Have you any concerns about statistical analyses in this paper?

No

Recommendation?

Major revision is needed (please make suggestions in comments)

Comments to the Author(s)

The authors propose an interesting new concept of validation tackling measuring errors and (known) uncertainties in the data. However, some unclear points remain which must be resolved before a possible publication.

1. Obviously, there are several concepts of validation. However, the most often used concept is not mentioned: "1. split a data set, calibrate the model (estimate its parameters) to one subset, and test the model with the estimated parameters on the other subset; 2. repeat with other splits; 3. compare calibration and validation metrics, i.e., the fitting error with the prediction error" This is, e.g., described in Chapter 16.4 of "Traffic Flow Dynamics" (Springer, 2013). Please discuss this concept. (Even if this paper is about testing "black box" software such as proprietary FEM packages, there surely are some parameters which the user can tune in order to do a proper calibration)

2. The authors base their metric on the mean absolute percentage error (MAPE). I wonder why they do not use the conventional calibration objective functions / validation metrics which are given in terms of the root-mean square percentage error (RMSPE). In my view, the VM can also be defined in terms of RMSPE. Moreover, calibration (finding the best-fit model parameters) and validation (testing the model with new data) are related and often based on the same goodness-of-fit function (GoF) which generally is some variant of RMSE. The authors should motivate their choice and discuss the relation to calibration

Minor

(a) Eq. (7) is inconsistent or, at least, uses an unconventional notation. It says there that $w_i = (w_k < e_{th})$ which is a Boolean. However, Booleans (true, false) cannot be added arithmetically. Sometimes, it is also interpreted as an indicator function with 1 if the boolean value=true, and zero, otherwise. However, what is really meant is

$$VM = \sum_i w_i I_{\{w_k < e_{th}\}} \quad (7)$$

where the indicator function $I_{\{w_k < e_{th}\}} = 1$ if the argument is true, and zero, otherwise.

(b) I do not see why fitting a distribution function should have a connection to Pascal's theorem which states something about conics while the typically S-shaped distribution functions are no element of the class of conics. For a known distribution, one needs one point more as there are degrees of freedom or parameters (e.g. theoretically only 3 points to fit a Gaussian, practically, much more). For an unknown distribution, nothing definite can be said.

(c) Fig. 5: The labels in the three scatter plots in the lower row are too small

Review form: Reviewer 2

Is the manuscript scientifically sound in its present form?

Yes

Are the interpretations and conclusions justified by the results?

Yes

Is the language acceptable?

Yes

Is it clear how to access all supporting data?

No

Do you have any ethical concerns with this paper?

No

Have you any concerns about statistical analyses in this paper?

No

Recommendation?

Accept with minor revision (please list in comments)

Comments to the Author(s)

General praise:

1. The development of VM represents a clear advance on the state of the art.
2. The claims surrounding VM are clear and not overstated.
3. The limitations are clearly and honestly discussed.
4. Overall, it was easy to understand the authors' method and contribution.

General criticism:

1. What about larger, more complex simulation validations? The three examples you cite are simple mechanical systems. My research domain is computational biology, where I simulate large dynamical systems that produce voluminous time series data that are difficult if not impossible to meaningfully validate against single-cell or even cell population time series measurements. While your VM and the methods you employ to evaluate it may not serve everyone's needs, please discuss the scope of its feasibility in a larger universe of data types.
2. In the discussion section, you address the assumption of experiment equaling reality. This is necessary. It's worth considering – without necessarily delving into a disquisition of the philosophy of science – an experiment is based on a mental model of reality, and this, in turn, is framed by the prevailing paradigm. So naturally, the experiment is quite removed from reality.
3. Your method depends on orthogonal decomposition into a pair of feature vectors, s_P and s_M , which have a nice one-to-one correspondence between their elements. In my experience, this is seldom the case; rather, experimental and simulation data fields are amenable to quite different methods of dimensionality reduction, the products of which don't neatly align. Please discuss this limitation.
4. Related to (3), the three examples you cite cleave neatly to stereoscopic digital image correlation systems that yield data fields amenable to orthogonal decomposition. Accordingly,

this is a bit conspicuous; the examples strike me as low-hanging fruit. In light of this, please discuss the application scope and biases of your methods more explicitly.

Specific comments:

Page 5, lines 25-32: One could use an uninformed, naive prior, derived from theory; this is objective.

Page 6, lines 34-44: Although this is clear, it could be developed a bit in a subsection, since it represents an important limitation.

Minor corrections or typos:

Page 2, line 29: "discussions about validation" => "discussions about algorithmic validation"

Page 2, line 42: "The concept of validation" => "The concept of formal validation"

Page 4, line 2: "where S_P and S_M" => "where s_P and s_M" (decide whether S is lower case, as in Eqn. 2, or upper case, as in this line, and be consistent)

Page 4, line 2: "measured data field" => "measured data fields,"

Page 5, Eqn. 4: it would be clearer if you wrote it like this: $e_k = | \frac{(s_P)_k - (s_M)_k}{\max\{m \text{ in } s_M\} | m |} |$

Page 6, line 20: "the weighted errors, w_k were" => "the weighted errors, w_k, were"

Page 6, lines 20-21: "the error threshold, e_{th} and" => "the error threshold, e_{th}, and"

Page 6, line 42: "associate with both" => "associated with both"

Page 7, line 52: "the relative error metric advances" => "the relative error metric improves upon"

Page 8, lines 34-37: I'm a fan of the Oxford comma. Also: "The values of the metric, VM for" => "The values of the metric, VM, for" or "The values of the metric VM for"

Page 8, line 44: "value of validation metric is very low" => "value of VM is very low"

Page 10, line 26: "above in italics, nevertheless" => "above in italics,; nevertheless"

Page 10, line 30: "The new validation metric, VM in equation (7) has been" => "The new validation metric, VM in equation (7), has been"

Page 11, line 17: "in the context of computational biology have" => "in the context of computational biology, have"

Page 11, line 27: "validation process address" => "validation process addresses"

Page 11, line 35: "decomposition technique then" => "decomposition technique, then"

Page 11, line 47: "static analysis and for a" => "static analysis, and for a"

Decision letter (RSOS-180687.R0)

03-Aug-2018

Dear Dr Patterson,

The editors assigned to your paper ("A Probabilistic Metric for the Validation of Computational Models") have now received comments from reviewers. We would like you to revise your paper in accordance with the referee and Associate Editor suggestions which can be found below (not including confidential reports to the Editor). Please note this decision does not guarantee eventual acceptance.

Please submit a copy of your revised paper before 26-Aug-2018. Please note that the revision deadline will expire at 00.00am on this date. If we do not hear from you within this time then it will be assumed that the paper has been withdrawn. In exceptional circumstances, extensions

may be possible if agreed with the Editorial Office in advance. We do not allow multiple rounds of revision so we urge you to make every effort to fully address all of the comments at this stage. If deemed necessary by the Editors, your manuscript will be sent back to one or more of the original reviewers for assessment. If the original reviewers are not available, we may invite new reviewers.

- Data accessibility

If you wish to submit your supporting data or code to Dryad (<http://datadryad.org/>), or modify your current submission to dryad, please use the following link:
<http://datadryad.org/submit?journalID=RSOS&manu=RSOS-180687>

- Competing interests

- Authors' contributions

- Acknowledgements

- Funding statement

Please note that Royal Society Open Science charge article processing charges for all new submissions that are accepted for publication. Charges will also apply to papers transferred to Royal Society Open Science from other Royal Society Publishing journals, as well as papers submitted as part of our collaboration with the Royal Society of Chemistry (<http://rsos.royalsocietypublishing.org/chemistry>). If your manuscript is newly submitted and subsequently accepted for publication, you will be asked to pay the article processing charge, unless you request a waiver and this is approved by Royal Society Publishing. You can find out more about the charges at <http://rsos.royalsocietypublishing.org/page/charges>. Should you have any queries, please contact openscience@royalsociety.org.

Kind regards,
Andrew Dunn
Senior Publishing Editor
Royal Society Open Science Editorial Office
Royal Society Open Science
openscience@royalsociety.org

on behalf of Prof Marta Kwiatkowska (Subject Editor)
openscience@royalsociety.org

Comments to Author:

Reviewers' Comments to Author:

Reviewer: 1

Comments to the Author(s)

The authors propose an interesting new concept of validation tackling measuring errors and (known) uncertainties in the data. However, some unclear points remain which must be resolved before a possible publication.

1. Obviously, there are several concepts of validation. However, the most often used concept is not mentioned: "1. split a data set, calibrate the model (estimate its parameters) to one subset, and test the model with the estimated parameters on the other subset; 2. repeat with other splits; 3. compare calibration and validation metrics, i.e., the fitting error with the prediction error" This is, e.g., described in Chapter 16.4 of "Traffic Flow Dynamics" (Springer, 2013). Please discuss this concept. (Even if this paper is about testing "black box" software such as proprietary FEM packages, there surely are some parameters which the user can tune in order to do a proper calibration)

2. The authors base their metric on the mean absolute percentage error (MAPE). I wonder why they do not use the conventional calibration objective functions / validation metrics which are given in terms of the root-mean square percentage error (RMSPE). In my view, the VM can also be defined in terms of RMSPE. Moreover, calibration (finding the best-fit model parameters) and validation (testing the model with new data) are related and often based on the same goodness-of-fit function (GoF) which generally is some variant of RMSE. The authors should motivate their choice and discuss the relation to calibration

Minor

(a) Eq. (7) is inconsistent or, at least, uses an unconventional notation. It says there that $w_i = (w_k < e_{th})$ which is a Boolean. However, Booleans (true, false) cannot be added arithmetically. Sometimes, it is also interpreted as an indicator function with 1 if the boolean value=true, and zero, otherwise. However, what is really meant is

$$VM = \sum_i w_i I_{\{w_k < e_{th}\}} \quad (7)$$

where the indicator function $I_{\{w_k < e_{th}\}} = 1$ if the argument is true, and zero, otherwise.

(b) I do not see why fitting a distribution function should have a connection to Pascal's theorem which states something about conics while the typically S-shaped distribution functions are no element of the class of conics. For a known distribution, one needs one point more as there are degrees of freedom or parameters (e.g. theoretically only 3 points to fit a Gaussian, practically, much more). For an unknown distribution, nothing definite can be said.

(c) Fig. 5: The labels in the three scatter plots in the lower row are too small

Reviewer: 2

Comments to the Author(s)

General praise:

1. The development of VM represents a clear advance on the state of the art.
2. The claims surrounding VM are clear and not overstated.
3. The limitations are clearly and honestly discussed.
4. Overall, it was easy to understand the authors' method and contribution.

General criticism:

1. What about larger, more complex simulation validations? The three examples you cite are

simple mechanical systems. My research domain is computational biology, where I simulate large dynamical systems that produce voluminous time series data that are difficult if not impossible to meaningfully validate against single-cell or even cell population time series measurements. While your VM and the methods you employ to evaluate it may not serve everyone's needs, please discuss the scope of its feasibility in a larger universe of data types.

2. In the discussion section, you address the assumption of experiment equaling reality. This is necessary. It's worth considering – without necessarily delving into a disquisition of the philosophy of science – an experiment is based on a mental model of reality, and this, in turn, is framed by the prevailing paradigm. So naturally, the experiment is quite removed from reality.

3. Your method depends on orthogonal decomposition into a pair of feature vectors, s_P and s_M , which have a nice one-to-one correspondence between their elements. In my experience, this is seldom the case; rather, experimental and simulation data fields are amenable to quite different methods of dimensionality reduction, the products of which don't neatly align. Please discuss this limitation.

4. Related to (3), the three examples you cite cleave neatly to stereoscopic digital image correlation systems that yield data fields amenable to orthogonal decomposition. Accordingly, this is a bit conspicuous; the examples strike me as low-hanging fruit. In light of this, please discuss the application scope and biases of your methods more explicitly.

Specific comments:

Page 5, lines 25-32: One could use an uninformed, naive prior, derived from theory; this is objective.

Page 6, lines 34-44: Although this is clear, it could be developed a bit in a subsection, since it represents an important limitation.

Minor corrections or typos:

Page 2, line 29: "discussions about validation" => "discussions about algorithmic validation"

Page 2, line 42: "The concept of validation" => "The concept of formal validation"

Page 4, line 2: "where S_P and S_M " => "where s_P and s_M " (decide whether S is lower case, as in Eqn. 2, or upper case, as in this line, and be consistent)

Page 4, line 2: "measured data field" => "measured data fields,"

Page 5, Eqn. 4: it would be clearer if you wrote it like this: $e_k = | \frac{(s_P)_k - (s_M)_k}{\max\{m \text{ in } s_M\}} |$

Page 6, line 20: "the weighted errors, w_k were" => "the weighted errors, w_k , were"

Page 6, lines 20-21: "the error threshold, e_{th} and" => "the error threshold, e_{th} , and"

Page 6, line 42: "associate with both" => "associated with both"

Page 7, line 52: "the relative error metric advances" => "the relative error metric improves upon"

Page 8, lines 34-37: I'm a fan of the Oxford comma. Also: "The values of the metric, VM for" => "The values of the metric, VM, for" or "The values of the metric VM for"

Page 8, line 44: "value of validation metric is very low" => "value of VM is very low"

Page 10, line 26: "above in italics, nevertheless" => "above in italics, nevertheless"

Page 10, line 30: "The new validation metric, VM in equation (7) has been" => "The new validation metric, VM in equation (7), has been"

Page 11, line 17: "in the context of computational biology have" => "in the context of computational biology, have"

Page 11, line 27: "validation process address" => "validation process addresses"

Page 11, line 35: "decomposition technique then" => "decomposition technique, then"

Page 11, line 47: "static analysis and for a" => "static analysis, and for a"

Author's Response to Decision Letter for (RSOS-180687.R0)

See Appendix A.

RSOS-180687.R1 (Revision)

Review form: Reviewer 1 (Martin Treiber)

Is the manuscript scientifically sound in its present form?

Yes

Are the interpretations and conclusions justified by the results?

Yes

Is the language acceptable?

Yes

Is it clear how to access all supporting data?

Not Applicable

Do you have any ethical concerns with this paper?

No

Have you any concerns about statistical analyses in this paper?

No

Recommendation?

Accept as is

Comments to the Author(s)

The authors have revised their paper according to the suggestions of my first report. I do not have further comments.

Martin Treiber

TU Dresden, Germany

Review form: Reviewer 2

Is the manuscript scientifically sound in its present form?

Yes

Are the interpretations and conclusions justified by the results?

Yes

Is the language acceptable?

Yes

Is it clear how to access all supporting data?

Yes

Do you have any ethical concerns with this paper?

No

Have you any concerns about statistical analyses in this paper?

No

Recommendation?

Accept as is

Comments to the Author(s)

Thank you. Much improved.

Decision letter (RSOS-180687.R1)

01-Oct-2018

Dear Dr Patterson,

I am pleased to inform you that your manuscript entitled "A Probabilistic Metric for the Validation of Computational Models" is now accepted for publication in Royal Society Open Science.

Please send the Editorial Office your original Figure and Table files as soon as possible. We cannot proceed to publication without these.

Kind regards,

Andrew Dunn

on behalf of Prof Marta Kwiatkowska (Subject Editor)
openscience@royalsociety.org

Reviewer comments to Author:

Reviewer: 1

Comments to the Author(s)

The authors have revised their paper according to the suggestions of my first report. I do not have further comments.

Martin Treiber
TU Dresden, Germany

Reviewer: 2

Comments to the Author(s)

Thank you. Much improved.

Appendix A

RSOS-180687

A Probabilistic Metric for the Validation of Computational Models

Authors' responses to reviewers' comments

Reviewer: 1

The authors propose an interesting new concept of validation tackling measuring errors and (known) uncertainties in the data. However, some unclear points remain which must be resolved before a possible publication.

- Thank you for these positive comments. We have addressed the unclear points below and by changes to the text which we have highlighted.

1. Obviously, there are several concepts of validation. However, the most often used concept is not mentioned: "1. split a data set, calibrate the model (estimate its parameters) to one subset, and test the model with the estimated parameters on the other subset; 2. repeat with other splits; 3. compare calibration and validation metrics, i.e., the fitting error with the prediction error" This is, e.g., described in Chapter 16.4 of "Traffic Flow Dynamics" (Springer, 2013). Please discuss this concept. (Even if this paper is about testing "black box" software such as proprietary FEM packages, there surely are some parameters which the user can tune in order to do a proper calibration).

- Yes, this was an omission on our part. We have described this type of validation in the revised introduction. The type of models, for which the validation metric is designed, would tend to have multiple inputs and outputs and a large number of degrees of freedom; while the quantity of measured data is limited such that calibration and validation cannot be performed using the same set of data. We have also included additional discussion of these issues.

2. The authors base their metric on the mean absolute percentage error (MAPE). I wonder why they do not use the conventional calibration objective functions / validation metrics which are given in terms of the root-mean square percentage error (RMSPE). In my view, the VM can also be defined in terms of RMSPE. Moreover, calibration (finding the best-fit model parameters) and validation (testing the model with new data) are related and often based on the same goodness-of-fit function (GoF) which generally is some variant of RMSE. The authors should motivate their choice and discuss the relation to calibration.

- Yes, it would be possible to define a validation metric in terms of root-mean-square-percentage-error (RMPSE); however, it was decided to use the absolute percentage error following the earlier work of Kat and Els. We do not believe that this choice is related to calibration because there is not the intimate connection to calibration for the type of models considered in the manuscript. Our rationale was mentioned briefly when introducing the error threshold but has now been reinforced in the discussion section where the advantages are highlighted.

Minor

(a) Eq. (7) is inconsistent or, at least, uses an unconventional notation. It says there that $w_i=(w_k < e_{th})$ which is a Boolean. However, Booleans (true, false) cannot be added arithmetically.

Sometimes, it is also interpreted as an indicator function with 1 if the boolean value=true, and zero, otherwise. However, what is really meant is

$$VM = \sum_i w_i I_{\{w_k \leq e_{th}\}} \quad (7)$$

where the indicator function $I_{\{w_k \leq e_{th}\}} = 1$ if the argument is true, and zero, otherwise.

- Equation (7) has been rewritten using an indicator function, as suggested.

(b) I do not see why fitting a distribution function should have a connection to Pascal's theorem which states something about conics while the typically S-shaped distribution functions are no element of the class of conics. For a known distribution, one needs one point more as there are degrees of freedom or parameters (e.g. theoretically only 3 points to fit a Gaussian, practically, much more). For an unknown distribution, nothing definite can be said.

- The reviewer's last sentence summarises our conundrum: the distribution is unknown and hence nothing definite can be said. However, it is inappropriate to use the validation metric when the cumulative distribution is defined by a very small number of points and there are far simpler alternatives for this scenario. Hence, we attempted to provide some guidance on the minimum number of points required to adequately define the validation metric. We have rewritten this paragraph of the paper in an attempt to address this issue more clearly.

(c) Fig. 5: The labels in the three scatter plots in the lower row are too small.

- The font of the labels have been increased by 50% and formatted in bold.

Reviewer: 2

General praise:

1. The development of VM represents a clear advance on the state of the art.
2. The claims surrounding VM are clear and not overstated.
3. The limitations are clearly and honestly discussed.
4. Overall, it was easy to understand the authors' method and contribution.

- Thank you for these favourable comments. We have endeavoured to address your criticisms in our responses below and by changes to the manuscript which are highlighted.

General criticism:

1. What about larger, more complex simulation validations? The three examples you cite are simple mechanical systems. My research domain is computational biology, where I simulate large dynamical systems that produce voluminous time series data that are difficult if not impossible to meaningfully validate against single-cell or even cell population time series measurements. While your VM and the methods you employ to evaluate it may not serve everyone's needs, please discuss the scope of its feasibility in a larger universe of data types.

- The authors agree that in its current form the validation metric cannot be applied to all forms of simulation, particularly those based on probabilistic models and representing dynamic systems with solutions in state space. As requested, we have extended our discussion of the scope of the validation metric and identified its limitation in the context of a large universe of data types.

2. In the discussion section, you address the assumption of experiment equaling reality. This is necessary. It's worth considering — without necessarily delving into a disquisition of the philosophy of science — an experiment is based on a mental model of reality, and this, in turn, is framed by the prevailing paradigm. So naturally, the experiment is quite removed from reality.

➤ Yes, this is an important issue and we have extended our discussion of it.

3. Your method depends on orthogonal decomposition into a pair of feature vectors, s_P and s_M , which have a nice one-to-one correspondence between their elements. In my experience, this is seldom the case; rather, experimental and simulation data fields are amenable to quite different methods of dimensionality reduction, the products of which don't neatly align. Please discuss this limitation.

➤ The methodology for the application of orthogonal decomposition was specifically designed by Sebastian et al [16] to yield one-to-one correspondence in order to avoid the issue raised by the reviewer. While other approaches to decomposition could be used, they would have to be applied within the same methodology to ensure one-to-one correspondence. We have included discussion of this issue and the consequential limitation in the revised manuscript.

4. Related to (3), the three examples you cite cleave neatly to stereoscopic digital image correlation systems that yield data fields amenable to orthogonal decomposition. Accordingly, this is a bit conspicuous; the examples strike me as low-hanging fruit. In light of this, please discuss the application scope and biases of your methods more explicitly.

➤ Yes, we are engineers and our group is concerned with the integrity of engineering components and structures; so, our examples are drawn from prior published studies in which we have been involved and hence had access to both model and experiment data. Digital image correlation has become ubiquitous in engineering mechanics experiments; however, the decomposition technique was applied by treating the predicted and measured data fields as images of colour contour maps; and thus, it is agnostic about the source of the data. It has been applied to data fields from projection moire and thermoelastic stress analysis in engineering mechanics and could be applied to any data fields that can be represented by images. We have discussed this application scope in the revised manuscript and made changes to emphasise the decomposition of images of data fields.

Specific comments:

Page 5, lines 25-32: One could use an uninformed, naive prior, derived from theory; this is objective.

➤ This observation has been added to the text. Thank you for highlighting it.

Page 6, lines 34-44: Although this is clear, it could be developed a bit in a subsection, since it represents an important limitation.

➤ As requested, and also in response to a similar comment by reviewer #1, this section has been rewritten and expanded.

Minor corrections or typos:

Page 2, line 29: "discussions about validation" => "discussions about algorithmic validation"

➤ The sentence was rephrased as "discussions about computational model validation" rather than "algorithmic validation" as suggested because 'algorithm' implies the solution process only as opposed to the representation and prediction of reality.

Page 2, line 42: "The concept of validation" => "The concept of formal validation"

- We acknowledge the need to qualify 'validation' in this sentence, but we prefer to rephrase as "The concept of model validation".

Page 4, line 2: "where S_P and S_M" => "where s_P and s_M" (decide whether S is lower case, as in Eqn. 2, or upper case, as in this line, and be consistent)

- Thank you for highlighting the inconsistency. We have changed the notation in equation (2) to upper case to be consistent with the rest of the text.

Page 4, line 2: "measured data field" => "measured data fields,"

- Corrected as suggested.

Page 5, Eqn. 4: it would be clearer if you wrote it like this: $e_k = \left| \frac{(s_P)_k - (s_M)_k}{\max\{m \text{ in } s_M\}} \right|$

- We have changed the notation in the equation, though we are not entirely clear about the reviewer's suggestion.

Page 6, line 20: "the weighted errors, w_k were" => "the weighted errors, w_k, were"

- Correction made as suggested.

Page 6, lines 20-21: "the error threshold, e_{th} and" => "the error threshold, e_{th}, and"

- Correction made as suggested.

Page 6, line 42: "associate with both" => "associated with both"

- Thank you for highlighting typographical error. We have corrected it.

Page 7, line 52: "the relative error metric advances" => "the relative error metric improves upon"

- We have changed the text as suggested.

Page 8, lines 34-37: I'm a fan of the Oxford comma. Also: "The values of the metric, VM for" => "The values of the metric, VM, for" or "The values of the metric VM for"

- The punctuation has been changed, as suggested.

Page 8, line 44: "value of validation metric is very low" => "value of VM is very low"

- Correction made as suggested.

Page 10, line 26: "above in italics, nevertheless" => "above in italics,; nevertheless"

- The comma has been replaced by a semi-colon.

Page 10, line 30: "The new validation metric, VM in equation (7) has been" => "The new validation metric, VM in equation (7), has been"

- The punctuation has been changed, as suggested.

Page 11, line 17: "in the context of computational biology have" => "in the context of computational biology, have"

- A comma has been inserted, as suggested.

Page 11, line 27: "validation process address" => "validation process addresses"

- Correction made, as suggested.

Page 11, line 35: "decomposition technique then" => "decomposition technique, then"

- A comma has been inserted, as suggested.

Page 11, line 47: "static analysis and for a" => "static analysis, and for a"

- The punctuation has been corrected.